# A longitudinal study to assess the frequency and cost of antivascular endothelial therapy, and inequalities in access, in England between 2005 and 2015

William Hollingworth,[1] Tim Jones,[2] Barnaby C Reeves,[3] Tunde Peto[4]

[1]Department of Population Health Sciences, University of Bristol, Bristol, UK
[2]NIHR CLAHRC West, University Hospitals Bristol NHS Foundation Trust, Bristol, UK
[3]Clinical Trials Evaluation Unit, Department of Translational Sciences, University of Bristol, Bristol, UK
[4]School of Medicine, Dentistry and Biomedical Sciences, Queen's University Belfast, Belfast, UK

**Correspondence to**
Dr William Hollingworth;
william.hollingworth@bristol.ac.uk

## ABSTRACT

**Objectives** High-cost antivascular endothelial growth factor (anti-VEGF) medicines for eye disorders challenge ophthalmologists and policymakers to provide fair access for patients while minimising costs. We describe the growth in the use and costs of these medicines and measure inequalities in access.

**Design** Longitudinal study using Hospital Episode Statistics (2005/2006 to 2014/2015) and hospital prescribing cost reports (2008/2009 to 2015/2016). We used Poisson regression to estimate standardised rates and explore temporal and geographical variations.

**Setting** National Health Service (NHS) care in England.

**Population** Patients receiving anti-VEGF injections for age-related macular degeneration, diabetic macular oedema and other eye disorders.

**Interventions** Higher-cost drugs (ranibizumab or aflibercept) recommended by the National Institute for Health and Care Excellence or lower-cost drug (bevacizumab) not licensed for eye disorders.

**Main outcome measures** National procedure rates and variation between and within clinical commissioning groups (CCGs). Cost of ranibizumab and aflibercept prescribing.

**Results** Injection procedures increased by 215% between 2010/2011 and 2014/2015. In 2014/2015 there were 388 031 procedures (714 per 100 000). There is no evidence that the dramatic growth in rates is slowing down. Since 2010/2011 the estimated cost of ranibizumab and aflibercept increased by 247% to £447 million in 2015/2016, equivalent to the entire annual budget of a CCG. There are large inequalities in access; in 2014/2015 procedure rates in a 'high use' CCG were 9.08 times higher than in a 'low use' CCG. In the South-West of England there was twofold variation in injections per patient per year (range 2.9 to 5.9).

**Conclusions** The high and rising cost of anti-VEGF therapy affects the ability of the NHS to provide care for other patients. Current regulations encourage the increasing use of ranibizumab and aflibercept rather than bevacizumab, which evidence suggests is more cost-effective. NHS patients in England do not have equal access to the most cost-effective care.

### Strengths and limitations of this study

► Our study summarises new data on nationwide access to effective, but high-cost ophthalmology drugs.
► Using population denominators allows us to standardise our analysis for factors, such as age and socioeconomic deprivation, which will affect the need for care.
► The lack of a specific antivascular endothelial growth factor drug code in Hospital Episode Statistics is a limitation; however we observed similar temporal trends in two independent data sources.
► The confidential nature of drug discounts mean that our estimate of drug costs will be too high if commissioners have negotiated substantial discounts.
► Conversely, the exclusion of administration and monitoring costs make our estimates conservative.

## INTRODUCTION

Clinical commissioning groups (CCGs) in England are responsible for commissioning healthcare for a geographically defined local population. On average, CCGs cover a population of 270 000 and have a budget of £300 million to commission services including elective and unplanned hospital care, and community and mental health services. While CCGs have a degree of autonomy in the care they commission, they must work within national constraints. In particular, CCGs must commission new interventions which have been evaluated to be cost-effective by the National Institute for Health and Care Excellence (NICE) technology appraisal process. This presents a challenge as many of these interventions are expensive. A balance is needed in order to provide fair access for patients eligible for the new interventions without

disadvantaging other patients using existing health services.

One recent challenge for CCGs is to commission anti-vascular endothelial growth factor (anti-VEGF) medicines for patients with a range of eye conditions. Anti-VEGF intravitreal injections reduce new blood vessel growth and swelling.[1] Two drugs (ranibizumab and aflibercept) are licensed for use in neovascular age-related macular degeneration (nAMD), diabetic macular oedema (DMO) or retinal vein occlusion (RVO) and myopic choroidal neovascularisation (MCNV). Ranibizumab was first recommended by NICE as cost-effective for use in nAMD in 2008.[2] After an initial rejection in 2011, NICE also recommended the use of ranibizumab for DMO in 2013 based on new evidence and an undisclosed discount on price offered by the manufacturer.[3] Subsequent NICE recommendations in other eye conditions and for aflibercept have followed (table 1). Ranibizumab and aflibercept have a National Health Service (NHS) indicative price in the region of £550 and £800 per injection, respectively.[4] The high prevalence of these conditions, the indefinite need for multiple injections per patient, and the associated procedure and monitoring costs mean that economic burden is substantial.

In reaching their recommendations, NICE did not compare ranibizumab or aflibercept with a pre-existing drug, bevacizumab. Bevacizumab also inhibits VEGF and is closely related to ranibizumab; it is licensed as a treatment for certain types of cancer but not in ophthalmology. Bevacizumab can be divided into aliquots of smaller doses for ophthalmic therapy and is a much less costly alternative (approximately £50 to £100 per injection).[5] Meta-analysis has demonstrated that bevacizumab and ranibizumab result in similar visual acuity outcomes for patients with nAMD.[6] The CATT and IVAN randomised trials of ranibizumab versus bevacizumab in nAMD both demonstrated that the drugs resulted in similar improvements in visual acuity.[7 8] IVAN also found that ranibizumab produced little or no gain in quality-adjusted life years despite being substantially more expensive.[9] Individual patient data meta-analysis indicates that the risk of serious adverse events with ranibizumab and bevacizumab is similar in nAMD.[10] In DMO, the DRCR. net three-arm randomised trial found that aflibercept and ranibizumab were not cost-effective relative to bevacizumab despite evidence that aflibercept resulted in better visual outcomes at 1 years and 2 years, particularly in those with worse visual acuity at baseline.[11 12] In RVO, the SCORE2 randomised trial concluded that bevacizumab was non-inferior to aflibercept for visual acuity after 6 months of treatment.[13] In 2014, the Royal College of Ophthalmologists and others called on UK health regulators, including NICE, to urgently review the use of bevacizumab in nAMD.[14] However, in 2015 the Department of Health declined to support clinicians or commissioners using bevacizumab for nAMD, considering unlicensed use 'on ground of cost alone' as potentially illegal under European Union (EU) law,[15] despite political support for the use of bevacizumab for ophthalmic indications in other EU countries such as Italy.[16] The remit of the forthcoming NICE nAMD clinical guidelines preclude making recommendations about bevacizumab.[17]

The emergence of anti-VEGF medicines provides ophthalmologists and CCGs with a clinical and economic dilemma in both providing fair access for eligible patients to drugs recommended by NICE while minimising the opportunity costs for other members of the population. In this study, we describe the growth in the use and cost of anti-VEGF intravitreal injections in the English NHS since 2005/2006. We quantify the variation in patient access to anti-VEGF therapy across England and explore the potential causes of inequality of access within the South-West region of England.

## METHODS
### Identifying anti-VEGF injections
We used the Hospital Episode Statistics (HES) admitted patient care (APC) and outpatient (OP) data sets to identify anti-VEGF injections. Anti-VEGF injections are performed in OP treatment rooms, but, in earlier years, were often coded in the APC data set as day case procedures. HES is a routinely collected data set that records all episodes of care provided to NHS-funded and privately insured patients admitted (day case or inpatient) to NHS hospitals or seen at an NHS OP clinic.[18] HES also includes data on NHS-funded patients treated in independent sector hospitals. We extracted anonymised individual episode records on all episodes contained in the HES APC (2005/2006 to 2014/2015) and OP (2010/2011 to 2014/2015) data sets. Up to 24 clinical procedures per

**Table 1** National Institute for Health and Care Excellence technology appraisals (TA) for antivascular endothelial growth factor therapies

| Condition | Ranibizumab | Aflibercept |
|---|---|---|
| Neovascular age-related macular degeneration | Recommend 8/2008 (TA 155) | Recommend 7/2013 (TA 294) |
| Diabetic macular oedema | Reject 11/2011 (TA 237)<br>Recommend 2/2013 (TA 274) | Recommend 7/2015 (TA 346) |
| Retinal vein occlusion | Recommend 5/2013 (TA 294) | Recommend 2/2014 (TA 305) |
| Myopic choroidal neovascularisation | Recommend 11/2013 (TA 298) | Recommend 9/2016 (TA 409) |

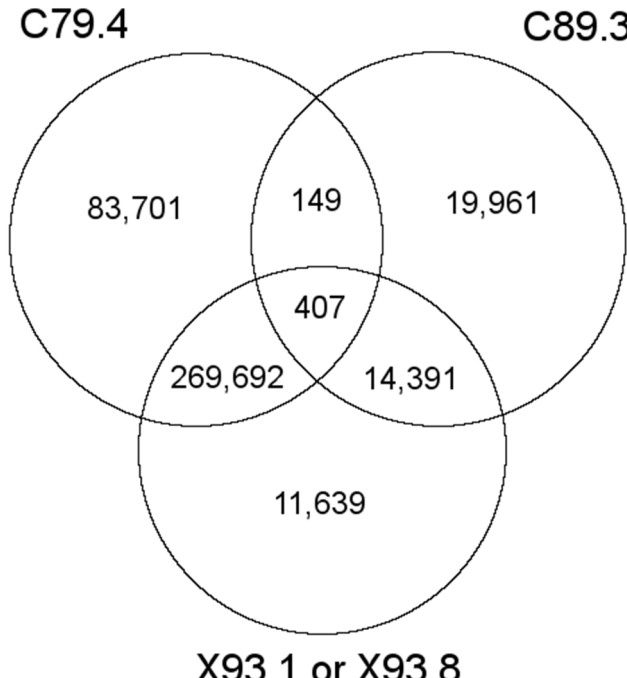

**Figure 1** Office of Population, Censuses and Surveys codes used to identify antivascular endothelial growth factor injections: 2014/2015. Note: 2479 eye injection procedures were excluded due to combination with X93.2 or X93.3 high-cost drug codes.

episode may be recorded using the Office of Population, Censuses and Surveys (OPCS) (fourth revision) codes.

Anti-VEGF injection procedures are not coded consistently. The majority of healthcare providers used the code 'injection into vitreous body, not elsewhere classified [C79.4]'. In preliminary exploration of HES data we looked for negative correlations between this procedure code and other ophthalmology procedure codes to identify potential alternate procedure codes. This identified that a substantial minority used a newer code 'Injection of therapeutic substance into posterior segment of eye, not elsewhere classified [C89.3]'. HES does not record the specific drug injected into the eye; however, since 2009 an OPCS code for high-cost subfoveal choroidal neovascularisation drugs band 1 (X93.1) has been in use. This code may be used for aflibercept, bevacizumab, ranibizumab and two other drugs rarely used in intravitreal injections, pegaptanib and verteporfin. We excluded any procedures using non anti-VEGF drugs (X93.2, X93.3).

In 2014/2015, 73.3% (284 490/388 031) of episodes including an eye injection procedure also coded a high-cost subfoveal choroidal neovascularisation drug. However, some hospitals performing hundreds of intravitreal injections each year never use this high-cost drug code. Conversely, 96% (284 490/296 129) of episodes coding a high-cost subfoveal choroidal neovascularisation drug also included an eye injection procedure (figure 1). In our analysis, we assumed that procedures coded as

C79.4 or C89.3 were anti-VEGF injections. We explore the importance of this assumption in our discussion.

A minority of patients receive bilateral injections on the same day, which may be coded using the supplemental 'bilateral operation' (Z94.1) OPCS code. When a patient had a bilateral procedure coded, we counted this as two injection procedures. When a patient had two injection procedures coded on the same day without any supplemental bilateral code, we assumed that this also represented bilateral procedures. If patients had more than two injection procedures coded on the same day, we excluded the 'excess' procedures as probable coding error duplicates (<0.03% of all procedures).

We used International Classification of Diseases 10th revision (ICD-10) diagnosis codes to identify patients with diabetes with ophthalmic complications (E10.3, E11.3, E12.3, E13.3 or E14.3) or diabetic retinopathy (H36.0) or with a primary code indicating 'other specified retinal disorders' (H35.8) combined with secondary codes indicating diabetes with ophthalmic complications as above; nAMD (H35.3); RVO (H34.8); and MCNV (H44.2), respectively. Diagnosis codes are well recorded for procedures in the HES APC data set, however in the HES OP data set diagnosis codes are often missing. Therefore, we limited our analysis of diagnostic indications to procedures recorded in the HES APC data set.

### Estimating procedure rates

National trends over time were estimated using directly standardised procedure rates[19] (per 100 000 population), with the population of England in 2014 as our standard population. We first summed the number of anti-VEGF injections, grouped by sex, quintiles of age and financial year. These procedure counts were used to calculate annual age-sex-specific rates, by dividing by the appropriate age-sex-specific midyear populations of England[20] (eg, for the 2012/2013 financial year, the mid-2012 populations were used). We weighted the annual age-sex-specific rates according to the population distribution of England in 2014, to produce directly standardised rates for each year.

Patients typically have more than one injection procedure in a year. Variation in patient access to anti-VEGF injections within England might be evident in both the number of patients treated and the number of procedures performed. Therefore, we estimated indirectly standardised rates[19] per 100 000 population for both patients and procedures. We first calculated age-sex-specific rates for England in 2014/2015, then multiplied these rates by the age-sex-specific population for the area of interest[20–22] (eg, CCG) and summed the results. This produced the expected number of patients and procedures for that area, if it were to have the same age-sex-specific rates as England. The expected number was then compared with the observed number of patients and procedures for that area. A Poisson regression model was fitted to the observed counts, with the expected counts as an offset and socioeconomic deprivation (using the overall score

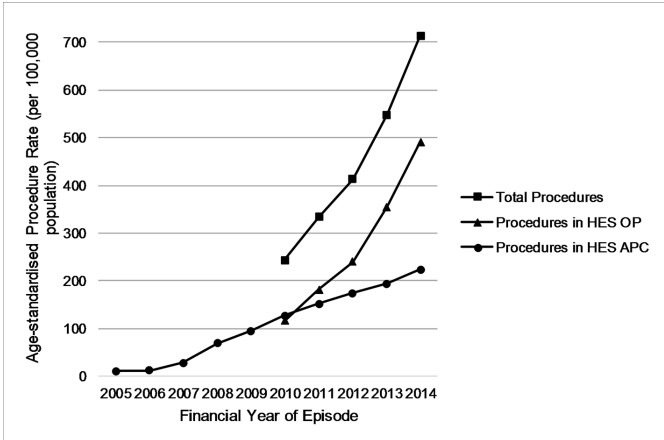

**Figure 2** Directly standardised intravitreal injection procedure rates, England, 2005/2006–2014/2015. APC, admitted patients care; HES, Hospital Episode Staristics; OP, outpatient.

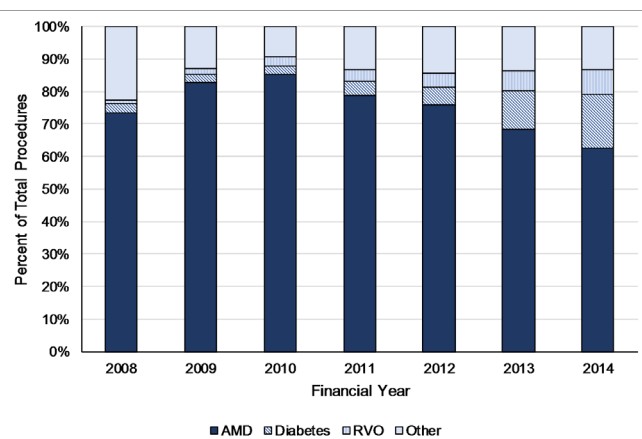

**Figure 3** Percentage of total eye injection procedures in Hospital Episode Statistics admitted patients care for different primary diagnoses, England, 2008/2009–2014/2015. AMD, age-related macular degeneration; RVO, retinal vein occlusion; due to very small numbers of procedures for myopic choroidal neovascularisation (≤10 per year), this was included in 'Other'.

from the English Indices of Multiple Deprivation[23]) and ethnicity (% white British[24]) as predictive factors. The model was then used to predict new expected counts for each area based on deprivation and ethnicity, and form indirectly standardised patient and procedure ratios (observed/expected). Statistical analyses were conducted using Stata/MP V.14.2 for Windows and we mapped variation in patient and procedure rates across England in 2014/2015 using ArcGIS ArcMap V.10.3.1 for Desktop.

### Exploring variation in access in the South-West region of England

For each CCG in the South-West region of England we estimated the median age and gender proportion in order to explore any differences in the demographic profile of patients receiving anti-VEGF injections. We estimated the annual number of procedures per patient in 2014/2015 to determine whether CCGs with high patient rates also had a higher number of procedures per patient. We also calculated and mapped indirectly standardised procedure ratios between Middle Layer Super Output Areas to explore whether CCG boundaries or hospital catchment areas were more strongly associated with access to anti-VEGF injections.

### Estimating the cost of anti-VEGF injections

We extracted data from NHS Digital Hospital prescribing costs reports[25] in order to estimate the costs of ranibizumab and aflibercept since 2008/2009 when ranibizumab was first approved by NICE. The NHS digital report summarises data collected from English NHS hospital pharmacies by the IMS Health Hospital Pharmacy Audit Index (HPAI). This includes high-cost medicines supplied to wards, OP clinics and satellite sites. Costs are reported in the HPAI and are calculated by IMS Health using standard price lists and therefore do not necessarily reflect any discounted prices paid by hospitals. Anti-VEGF therapy also incurs administration and monitoring costs; however, the IVAN trial indicated that, in the ranibizumab

group, drug costs accounted for 80%–88% of total costs.[9] Injection frequency and monitoring visits may be lower for aflibercept where fixed 2-monthly interval dosing is recommended after an initial period of monthly loading doses.

Bevacizumab is not included in HPAI and due to the lack of drug coding in HES data it is impossible to accurately estimate the current use of bevacizumab for eye conditions. A study in 2015 based on 189 Freedom of Information requests across the UK estimated the relative prevalence of injections of ranibizumab, aflibercept and bevacizumab to be 61.1%, 36.0% and 2.8%, respectively.[26]

## RESULTS
### The use and cost of anti-VEGF intravitreal injections

In 2014/2015 there were 388 031 intravitreal injection procedures, equivalent to 714 procedures per 100 000 population (figure 2). This compares to 123 006 procedures in 2010/2011 and represents a 215% increase in the number of procedures over the 5-year period. Before 2007, intravitreal injections were rarely used. The rise in procedure rates since then is dramatic and there are no signs that it is levelling off.

Among episodes recorded in the HES APC data set, the majority (62% in 2014/2015) had diagnosis codes indicating nAMD (figure 3). The relative use of diagnosis codes indicating DMO or RVO show small recent increases associated with NICE approvals for these indications in 2013. The estimated total cost of hospital prescribing of ranibizumab and aflibercept increased by 197% from £129 million to £383 million in the 5 years between 2010/2011 and 2014/2015 (figure 4). The estimated total cost of ranibizumab and aflibercept had risen to £447 million in 2015/2016, predominantly due to the increasing use of aflibercept.

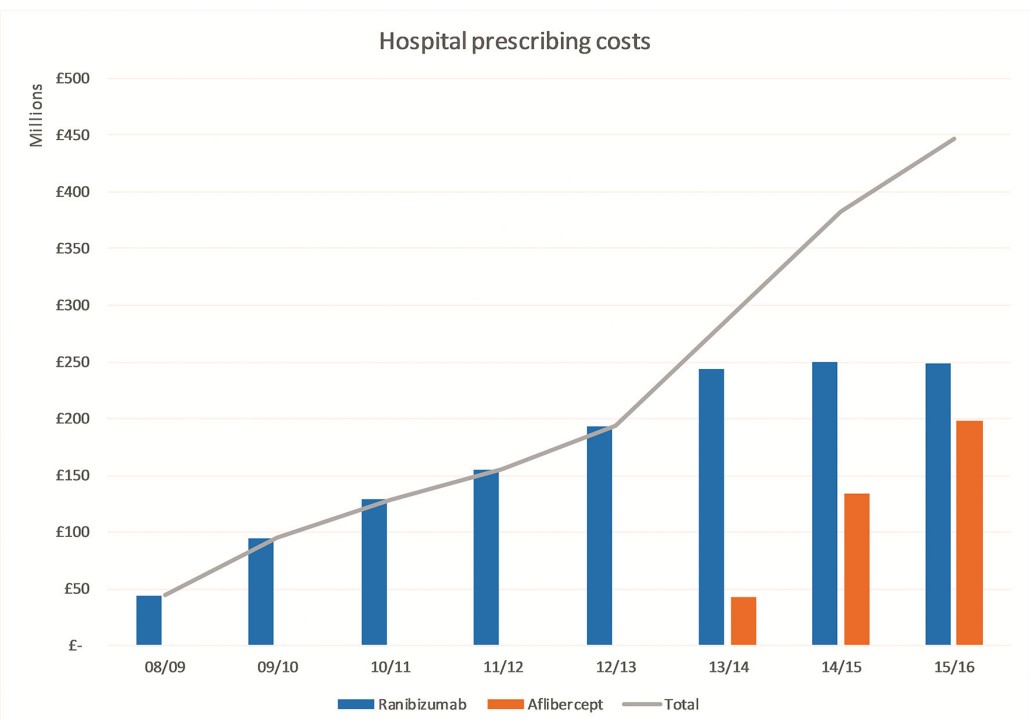

**Figure 4** Hospital prescribing costs for ranibizumab and aflibercept, 2008/2009 to 2015/2016.

## Variation in patient access to anti-VEGF therapy

In 2014/2015 there was substantial variation in procedure rates between CCGs, even after adjusting for age, sex, deprivation and ethnicity profiles (figure 5A). The map demonstrates pockets of very high use (>150% times the expected rate), for example, in the Bristol area and in parts of North London and Norfolk. There are also areas where procedure rates are less than 50% of the expected

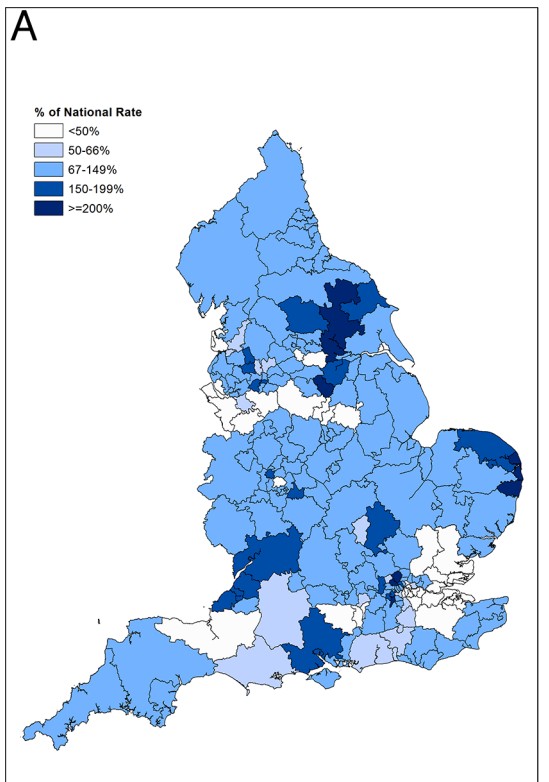
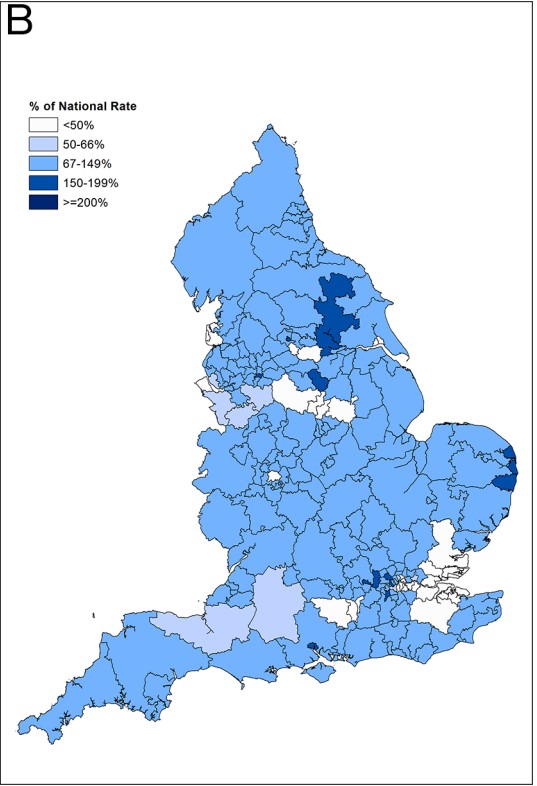

**Figure 5** (A) Indirectly age-sex-deprivation-ethnicity standardised procedure ratios for combined eye injection procedures (OPCS-4: C794 and C893) by CCG in England, 2014/2015. (B) Indirectly age-sex-deprivation-ethnicity standardised patient ratios for combined eye injection procedures (OPCS-4: C794 and C893) by CCG in England, 2014/2015. CCG, clinical commissioning group; OPCS, Office of Population, Censuses and Surveys.

rate, for example, in parts of Essex. There was less pronounced, but still substantial variation in patient rates between CCGs, after adjusting for age, sex, deprivation and ethnicity profiles (figure 5B). Areas of the country with high procedure rates also tended to have higher proportions of their populations having treatment. In 2014/2015 the ratio in patient rates between a 'high use' CCG at the 90th centile and a 'low use' CCG at the 10th centile was 4.98. This ratio has only slightly decreased since 2010/2011, when the ratio was 6.07, indicating that equality of access to care is only slowly improving over time.

Examining the South-West region of England in more detail (figure 6A,B), it is evident that there is substantial variation in access to anti-VEGF injections within CCGs. For example, in both Somerset and Wiltshire there are areas of relatively high procedure rates closer to the major conurbations of Bath, Bristol and Swindon. Procedure rates are much lower than expected in other areas of these CCGs, even though these other areas are served by local hospitals. The age and gender profiles of patients receiving anti-VEGF therapy were similar between CCGs (table 2). Median age ranged from 77 years to 82 years and between 39% and 49% of patients were male. There was twofold variation in the number of procedures per patient per year ranging from 2.9 in Cornwall to 5.9 in South Gloucestershire. The standardised CCG patient rate was strongly correlated with the number of procedures per patient (Spearman's ρ=0.71, p<0.001).

## DISCUSSION
### Statement of principal findings
There is no evidence that the dramatic growth in the use of high-cost anti-VEGF injections in ophthalmology is slowing down. Injection procedures increased by 215% between 2010/2011 and 2014/2015. There are large inequalities in access to anti-VEGF therapy. This inequality is not explained by differences in demographics and occurs both within and between CCGs. Procurement of high-cost anti-VEGF drugs has placed a large and increasing strain on the NHS. In 2015/2016, the NHS spent an estimated £447 million on drug costs alone, equivalent to the entire annual budget of a large CCG.

### Strengths and weaknesses of the study
Our study summarises new data on nationwide access to effective but high-cost ophthalmology drugs. Using population denominators allows us to standardise our analysis for factors, such as age and socioeconomic deprivation, which will affect the need for care. HES data record patients' area of residence, so we compare access based on place of residence rather than place of treatment. The combination of HES APC and OP data sets is essential in order to accurately monitor intravitreal injections over time. As reimbursement payments depend on procedure coding, there is an incentive for providers to fully code procedures. However, the lack of a specific anti-VEGF drug code in HES

is a limitation. Other similar procedures such as the intravitreal injection of dexamethasone implants have distinct OPCS procedure codes, although we cannot exclude the possibility of miscoding. We observed similar temporal trends in two independent data sources measuring procedures (ie, HES) and drug costs (ie, HPAI), which provides reassurance about the validity of our time-trend analyses. Previous work has demonstrated that HES generally has good sensitivity in identifying new procedures, but may lack specificity, particularly in the absence of a single specific procedure code.[27] It is plausible that drug price discounts account for some of the observed variations, if CCGs who have larger discounts offer more therapy. The confidential nature of discounts means that our estimate of drug costs will be too high if CCGs have negotiated substantial discounts. Conversely, the exclusion of administration and monitoring costs makes our estimates conservative and there is a substantial economic burden for patients. Travel and postappointment recovery time have a large impact on patients and carers.[28] The costs of all three anti-VEGF drugs will be offset, to a greater or lesser extent, in the longer term by reduced need for NHS and social services related to vision loss.

### Comparison with other studies
Between 1989/1990 and 2008/2009, Keenan et al observed an increase in intravitreal injection rates from 0.4 per 100 000 to 59.5 per 100.000 with wide variations in access between different local authorities in England.[29] Our data demonstrate that this rapid growth has continued and even accelerated in the following 6 years. We were also able to explore variation in access to care at a more granular level and demonstrate that access varies markedly both within and between CCGs, suggesting that variations are likely to be due, at least in part, to the availability of trained staff (eg, ophthalmologists and nurse practitioners) and facilities in local hospitals rather than CCG-wide policies. Using insurance data, Parikh et al observed a continually increasing trend in anti-VEGF use in the USA between 2006 and 2015.[30] In line with our study, Parikh et al also found that anti-VEGF drug use was predominantly for nAMD. It is impossible to directly compare England and US rates as the denominator used in the Parikh et al study excludes patients without commercial or Medicare Advantage insurance. However, anti-VEGF therapy rates in the USA and UK appear to be of the same order of magnitude. Specific coding of anti-VEGF drugs allowed Parikh et al to document the rise in the use of aflibercept (28% of all anti-VEGF injections in 2015) and the concurrent levelling off in the use of ranibizumab (17%) and bevacizumab (55%). This is in stark contrast to England where ranibizumab and aflibercept predominate. This is likely to be a result of legal challenges to the use of bevacizumab and the lack of Department of Health support for commissioners.[31]

### Implications for policymakers and clinicians
Based on the IVAN trial, Dakin et al estimated that the NHS could save £102 million per year by switching from

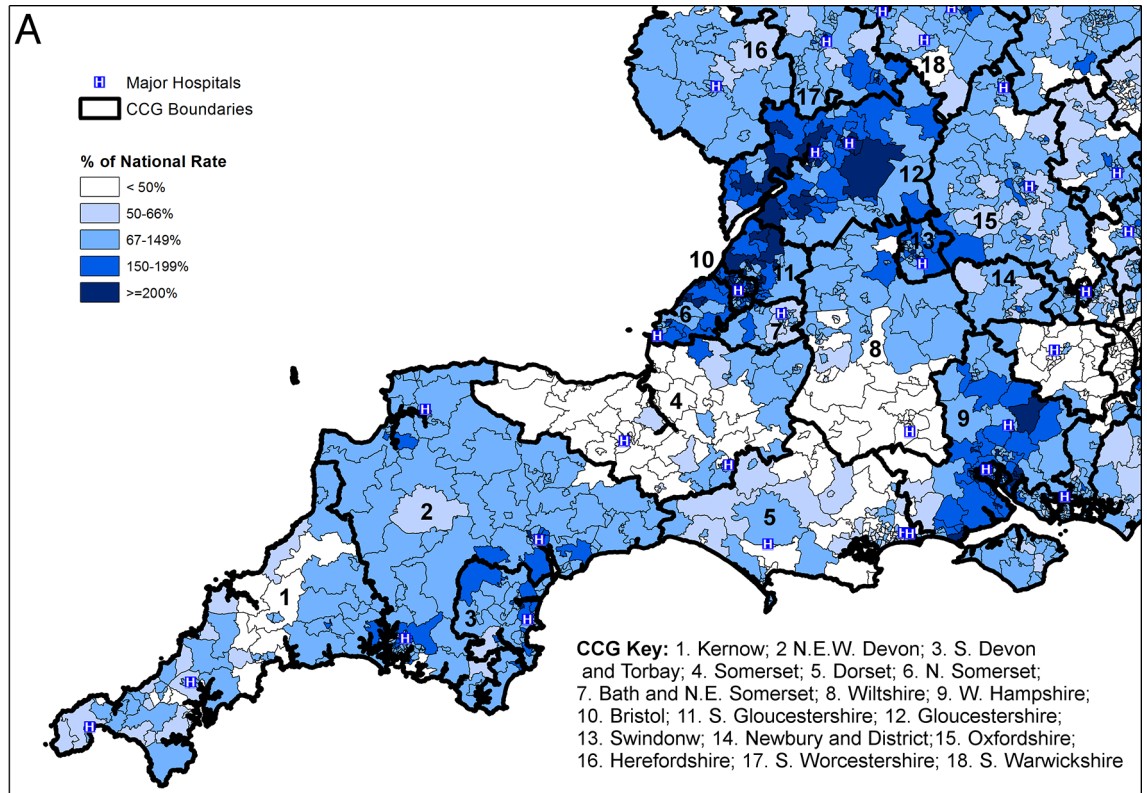

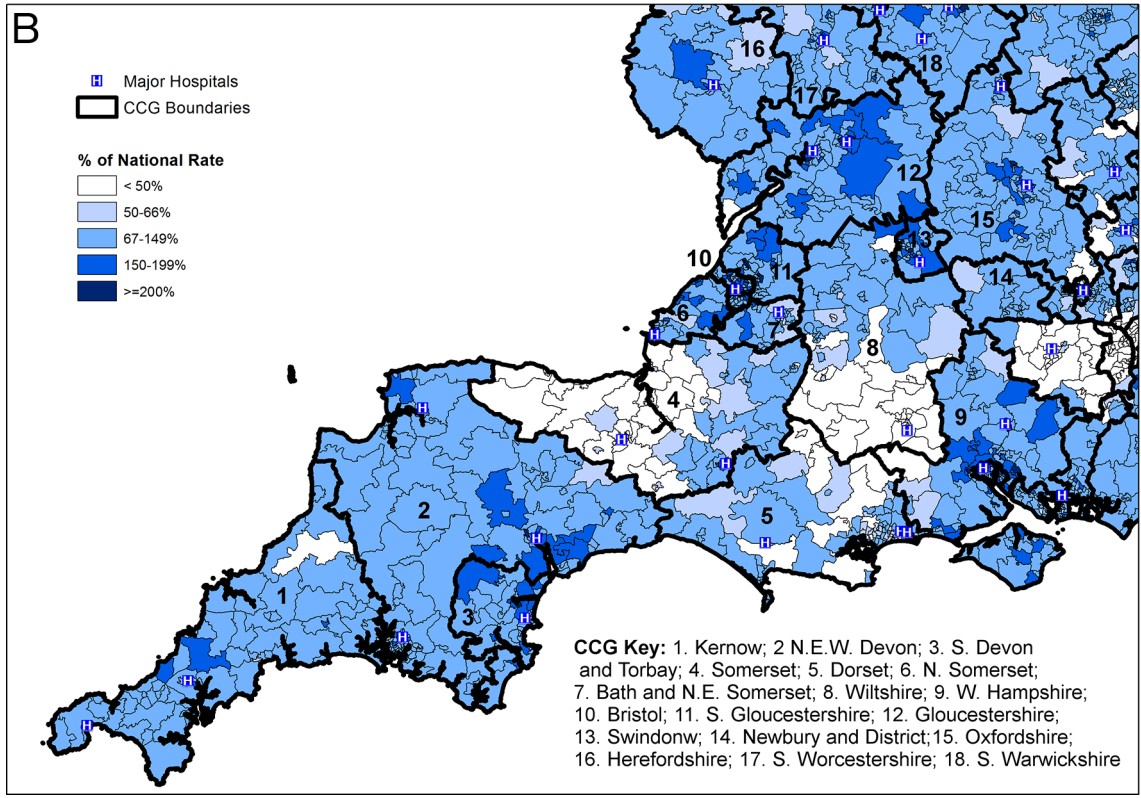

**Figure 6** (A) Indirectly standardised procedure ratios for combined eye injection procedures by Middle Layer Super Output Areas (MSOA) in SW England, 2014/2015. (B) Indirectly standardised patient ratios for combined eye injection procedures by MSOA in SW England, 2014/2015. Note: Major hospitals marked including those in National Health Service (NHS) Trusts providing 100+ procedures. CCG, clinical commissioning group.

**Table 2** Demographics of anti-VEGF procedures in South-West CCGs, 2014/2015; ordered by procedures per patient

| CCG Name | % Male | Median age | Procedures per patient (95% CI) | Procedure rate* (95% CI†) | Patient rate* (95% CI) |
|---|---|---|---|---|---|
| NHS Kernow CCG | 43 | 79 | 2.86 (2.75 to 2.97) | 522 (490 to 556) | 172 (191 to 104) |
| NHS Dorset CCG | 43 | 82 | 3.32 (3.21 to 3.43) | 419 (394 to 445) | 122 (136 to 74) |
| NHS Herefordshire CCG | 38 | 80 | 3.34 (3.14 to 3.53) | 631 (569 to 700) | 175 (210 to 110) |
| NHS Somerset CCG | 45 | 78 | 3.48 (3.27 to 3.68) | 304 (276 to 334) | 92 (106 to 57) |
| NHS South Warwickshire CCG | 44 | 79 | 3.63 (3.46 to 3.8) | 773 (708 to 844) | 206 (240 to 128) |
| NHS Oxfordshire CCG | 40 | 81 | 3.63 (3.5 to 3.76) | 723 (679 to 771) | 197 (220 to 120) |
| NHS Northern-Eastern and Western Devon CCG | 41 | 81 | 4.05 (3.95 to 4.16) | 830 (792 to 869) | 201 (218 to 121) |
| NHS Newbury and district CCG | 41 | 79 | 4.14 (3.72 to 4.55) | 637 (530 to 764) | 136 (186 to 92) |
| NHS Wiltshire CCG | 46 | 79 | 4.17 (3.95 to 4.39) | 459 (419 to 502) | 103 (121 to 64) |
| NHS South Devon and Torbay CCG | 40 | 81 | 4.2 (4.05 to 4.35) | 923 (860 to 991) | 206 (235 to 127) |
| NHS South Worcestershire CCG | 42 | 79 | 4.36 (4.16 to 4.56) | 875 (806 to 950) | 188 (217 to 116) |
| NHS Bath and North-East Somerset CCG | 49 | 81 | 4.44 (4.14 to 4.73) | 735 (650 to 831) | 152 (189 to 97) |
| NHS Swindon CCG | 44 | 77 | 4.54 (4.26 to 4.82) | 1034 (928 to 1153) | 216 (260 to 137) |
| NHS West Hampshire CCG | 42 | 80 | 5 (4.85 to 5.15) | 1125 (1063 to 1190) | 221 (244 to 133) |
| NHS North Somerset CCG | 43 | 80 | 5.43 (5.15 to 5.71) | 1236 (1130 to 1352) | 217 (253 to 135) |
| NHS Bristol CCG | 43 | 80 | 5.49 (5.26 to 5.72) | 1346 (1248 to 1451) | 237 (270 to 146) |
| NHS Gloucestershire CCG | 42 | 81 | 5.51 (5.33 to 5.68) | 1263 (1194 to 1336) | 226 (249 to 137) |
| NHS South Gloucestershire CCG | 46 | 79 | 5.93 (5.65 to 6.22) | 1379 (1264 to 1505) | 222 (258 to 138) |

* Procedure rate and patient rate are the indirectly standardised rates per 100 000 population.
†95% CIs on the procedure rate are calculated using Stukel *et al*'s method for recurrent outcomes.[40]
anti-VEGF, antivascular endothelial growth factor; CCG, clinical commissioning group; NHS, National Health Service.

ranibizumab to bevacizumab.[9] In the USA, Hutton *et al* estimated that a similar switch would save Medicare $18 billion over a 10-year period.[32] Given the lack of political and regulatory support for clinicians to use bevacizumab, it is unsurprising that most in England do not use it. This has led to the very unusual situation whereby NHS is paying for more expensive therapy than healthcare insurers in the USA. The patent on ranibizumab is due to expire in Europe in 2022;[33] in the meantime, based on current trends, NHS may spend billions on anti-VEGF injections, before biosimilar drugs become available. Our data, demonstrating large variations between CCGs in the numbers of patients accessing anti-VEGF services, also suggest that there is considerable potential unmet need in some areas of the country. Given that many CCGs are already in financial deficit, most will struggle to afford to treat more patients unless they are able to switch to bevacizumab. The political decision not to support NHS use of bevacizumab in eye conditions is in stark contrast to decisions taken in other EU countries[16] and has very large negative consequences for NHS patients.

In April 2017, NICE introduced a budget impact test for new technologies under appraisal that are expected to cost more than £20 million per annum. This allows NHS England up to 3 years to conduct commercial negotiations on price rather than the current requirement to fund within 90 days.[34] This policy should strengthen the future negotiating power of the NHS for expensive drugs.

However, it does not address the fundamental problem that NICE has not included cost-effective but unlicensed drugs in technology appraisals or clinical guideline recommendations.

The widespread variation in procedure rates that we observed might be due to travel distance affecting patient access to diagnostic and therapeutic services, particularly in more remote areas. Lengthy travel distance is associated with treatment discontinuance.[35] However, many of the low injection rate areas in the South-West of England are close to hospitals. Other unmeasured factors, such as the importance of vision for work or usual activities, may also be associated with patient uptake and compliance with anti-VEGF therapy. The frequency of injections per patient per year was lower than reported in trials of therapy initiation. The averages in our study will be drawn downwards due to patients starting and discontinuing therapy partway through the year and by patients who have been on anti-VEGF therapy for many years and need fewer injections.

### Unanswered questions and future research
Given the high cost of the drugs, it is imperative that clinical services are organised to deliver care efficiently. For example, nurse-administered injections,[36] treat and extend protocols,[37] optometrist-led monitoring[38] and stereotactic radiotherapy,[39] all offer the potential to reduce costs and require stronger evidence. The wide

variation between CCGs in the number of injections per patient represents a natural experiment. Observational studies could compare costs and outcomes between areas of the country with high and low injection frequencies. However, the lack of a nationwide register of patients receiving anti-VEGF injections and the lack of drug coding in HES severely limit the potential to conduct this research.

## CONCLUSIONS

Anti-VEGF therapy is effective in most cases and has improved the quality of life of many thousands of patients in England. However, current regulations encourage the increasing use of ranibizumab and aflibercept rather than bevacizumab, which, evidence suggests, is more cost-effective. This limits the ability of NHS to pay for care for other patients. NHS patients in England do not have equal access to the most cost-effective care.

**Acknowledgements** Hospital Episode Statistics were provided by NHS Digital under data sharing agreement (NIC-17875-X7K1V) with the University of Bristol. Copyright © 2016, re-used with the permission of The Health & Social Care Information Centre. All rights reserved.

**Contributors** WH initiated and designed the study, contributed to the data analysis and interpretation of data, and drafted and revised the paper. He is guarantor. TJ contributed to the study design, cleaned and analysed the data, contributed to the interpretation of data, and revised the draft paper. BCR and TP contributed to the interpretation of data and revised the draft paper. All authors approved the final version of the paper. All authors had full access to all of the data in the study and can take responsibility for the integrity of the data and the accuracy of the data analysis.

**Funding** WH and TJ receive funding from the National Institute for Health Research Collaboration for Leadership in Applied Health Research and Care (NIHR CLAHRC) West.

**Disclaimer** The views expressed are those of the authors and not necessarily those of the NHS, the NIHR or the Department of Health.

**Competing interests** None declared.

**Provenance and peer review** Not commissioned; externally peer reviewed.

**Data sharing statement** No additional data are available.

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
