## [Reviewer comments · BMJ Open]

ARTICLE DETAILS

TITLE (PROVISIONAL)	A longitudinal study to assess the frequency and cost of anti-vascular endothelial therapy, and inequalities in access, in England between 2005 and 2015.
AUTHORS	Hollingworth, William; Jones, Tim; Reeves, Barnaby; Peto, Tunde

VERSION 1 – REVIEW

REVIEWER	Ravi Parikh MD MPH Yale University School of Medicine USA
REVIEW RETURNED	25-Jun-2017

GENERAL COMMENTS	Overall interesting and important work although limited by lack of direct information on breakdown of medications. I think this paper should be accepted with the following revisions made. 1. When mentioning the IVAN trial, it is important to also mention the US Comparison of Age Related Macular Degeneration Treatment Trial (CATT), which first demonstrated similar efficacy of bevacizumab and ranibizumab for nAMD.2. Although somewhat mentioned, it is also important to mention explicitly that the only evidence that any of the drugs is superior is aflibercept to bevacizumab at 2 years for patients with DMO with 20/50 or worse vision based upon DRCR.net Protocol T (aflibercept was also found to be superior at 1 year to ranibizumab but that difference was statistically insignificant at year 2). Otherwise there is no comparative trial evidence showing that aflibercept is better than bevacizumab for RVO or nAMD. Further, for nAMD it maybe useful to also note that aflibercept was shown to be non-inferior to ranibizumab, which has been demonstrated to have equal efficacy to bevacizumab in CATT and IVAN.3. Can you elaborate more on the methods for evaluating cost of ranibizumab and aflibercept? Does the NHS report directly state the cost, and how does one determine the cost of aflibercept or ranibizumab without the drug codes for each injection?4. Page 9 lines 17-19 “Subsequent [...] bevacizumab,” please cite and provide an example of the subsequent legal challenges and place this portion in the discussion.5. Lines 9-11 on page 15 “The political decision [...] NHS patients” please cite some evidence of different nations support of bevacizumab use.
---

	6. In the conclusion, this paper allows us to draw the conclusions that a. anti-VEGF drug use is increasing, b. costs of ranibizumab and aflibercept are increasing, c this is wide variation on use of these medications throughout the UK, and that substituting bevacizumab would likely lead to relative savings compared to the status quo. Relative savings can be achieved if NICE recommended use of bevacizumab, however as anti-VEGF use increases overall costs may still rise. 7. In the conclusion, lines 14-17 should be reworded. Although I agree, and readers should be able to “connect the dots,” this study does not directly show how anti-VEGF costs limit care and how NICE is unable to provide equal access to cost effective care. I think it would be preferable that after point 6 is addressed to note that NICE’s inability to recommend bevacizumab, a more cost effective medication with proven efficacy, may lead to the inability to achieve equal access. Further, I do think it is accurate that in this case they are failing to encourage cost effective care. 8. The title should be changed to “the cost of turning a blind eye to unlicensed medicines: NICE recommendations on anti-vascular endothelial therapy.”
--	--

REVIEWER	Andrew F. Smith Dept. of Ophthalmology King's College London London, UK None
REVIEW RETURNED	21-Jul-2017

GENERAL COMMENTS	Overview: This is an important piece of research in a complex area, namely the prevention of vision loss and blindness due to ARMD in the UK and the cost implications of using expensive licensed medicines versus cheaper off-label drugs with similar efficacy and ensuring access by patients to these medications. The assumptions of the analysis, however, are somewhat limited and largely restricted to considerations of the impact on the NHS drug budget and inequalities in accessing anti-VEGF treatment across the UK. In my view, both central issues have been handled rather simplistically and would benefit from further elucidation of the wider issues at play. Proposed revisions: In particular the authors may wish to consider the following revisions which should yield a more nuanced treatment of this complex topic. 1) The authors have relied solely on HES statistics to explore trends in the use of licensed and off label anti-VEGF agents across the UK. Given the difficulties in precise coding, it might be worth obtaining access to IMS pharmaceutical sales data for the UK to conduct a parallel analysis by drug names to see if the same quantities of both the licensed and off label anti-VEGF medicines were sold during the same time periods explored and roughly match the analysis of the HES data sets.
---

2) Moreover, an analysis of actual UK pharmaceutical sales data should show the total quantities of both the licensed and off label anti-VEGF drugs sold in different parts of the UK and CCGs regions.

3) The authors may also conduct a short survey of CCGs about their purchasing of anti-VEGF agents to see if they received any discounts and whether those CCGs who extended discounts also accounted for higher numbers of anti-VEGF injections relative to CCGs paying full price.

4) No mention was made of perhaps the most important factor governing who has access to licensed or off-label anti-VEGF medicines, namely the supply and distribution of ophthalmologists as well as specially trained nurse practitioners across the UK. This could be achieved relatively easily by superimposing a map showing the distribution of ophthalmologists across the UK (particularly medical retinal specialists) and then correlating this distribution of eye care personnel to inequalities in access anti-VEGF agents. I suspect that accessing eye care for wet ARMD has much more to do with eye care personnel than the cost of the licensed anti-VEGF agents. The Royal College of Ophthalmologists would, I am sure, be able to assist with the provision of such workforce data. Alternatively, CCGs should have a list of the number of ophthalmologists by staff grade which could be used for such purposes.

5) The economic assumptions of the model make no allowance for the fact that the total societal costs of treating and managing patients with ARMD may well be on the decline, such that increased expenditures on licensed or off-label anti-VEGF agents owing to their efficacy, may actually be decreasing consumption on other healthcare or societal costs such as nursing home care, low vision rehabilitation services, and mobility aids, etc. for those who would have otherwise gone blind due to untreated wet ARMD. The authors might do well to calculate the cost of untreated vision loss and blindness due to wet ARMD versus the cost of treating wet ARMD patients either with the licensed or off-label anti-VEGF before concluding that the NHS is not getting value for money at least from a macro-economic or societal perspective.

6) Another crucial point worth exploring is the degree to which those capture in the HES dataset represent "true" candidates and that ophthalmologists are not over treating ARMD patients and "cashing in" on the anti-VEGF ARMD bonanza, especially when the consequences of no treatment are further vision loss and blindness to patients. In essence, how does one know that all ARMD patients receiving either the licensed or off-label anti-VEGF treatment meet the criteria and are truly medically warranted? Or could ophthalmologist simply be over treating ARMD patients in the UK for financial gain? This could be answered by examining the number of persons in the UK who would have had the treatable wet form of ARMD from epidemiologic studies and comparing this with the numbers actually treated in the UK during the same time period to see if an appropriate number were being treated or not. This could be achieved by making assumptions about the number of injections per patients and examining the total number of injections annually during the period of interest and comparing this with the expected epidemiological burden of wet ARMD in the UK over the same time period. We would then know if the ophthalmologist were pulling the wool over the eyes of the NHS!

	7) In connection with the above point (6), the authors might wish to consider the provision of a better means of monitoring who would benefit from anti-VEGF agents (licensed or off-label) and thereby cut down on the potential for abuses in the system, including, expanding the indications beyond the realm of sufficient clinical data. This in turn, might well attenuate the rising costs. 8) The authors further assume that the efficacy between Lucentis and Avastin are comparable. A review of reported adverse events may well reveal this not to be the case owing to basic pharmacological differences in the mechanism of action between the two anti-VEGF agents, with one having a larger molecular weight and longer half-life in the circulator system (the off-label Avastin) versus that with the smaller molecular weight and shorter half-life (the licensed anti-VEGF agent (Lucentis)). 9) Finally, the authors may wish to comment on the precedent that the Lucentis / Avastin saga has set for the role of Medicine Control Agencies, like the EMA and FDA who's primary concern remains the licensing of safe and effective medications for its populations, while concerns regarding costs and cost-effectiveness remain the preserve of other agencies, and seemingly secondary agencies, such as NICE in the UK or CMS in the USA for example.
--	--

VERSION 1 – AUTHOR RESPONSE

Reviewer: 1

Reviewer Name: Ravi Parikh MD MPH

Institution and Country: Yale University School of Medicine, USA Please state any competing interests: None declared

Please leave your comments for the authors below

Overall interesting and important work although limited by lack of direct information on breakdown of medications. I think this paper should be accepted with the following revisions made.

1. When mentioning the IVAN trial, it is important to also mention the US Comparison of Age Related Macular Degeneration Treatment Trial (CATT), which first demonstrated similar efficacy of bevacizumab and ranibizumab for nAMD.

Response:

We have revised this section to mention the findings of the CATT study.

2. Although somewhat mentioned, it is also important to mention explicitly that the only evidence that any of the drugs is superior is aflibercept to bevacizumab at 2 years for patients with DMO with 20/50 or worse vision based upon DRRCR.net Protocol T (aflibercept was also found to be superior at 1 year to ranibizumab but that difference was statistically insignificant at year 2). Otherwise there is no comparative trial evidence showing that aflibercept is better than bevacizumab for RVO or nAMD. Further, for nAMD it maybe useful to also note that aflibercept was shown to be non-inferior to ranibizumab, which has been demonstrated to have equal efficacy to bevacizumab in CATT and IVAN.

Response:

We have revised this section to explicitly state that the DRCCR.net trial found that aflibercept resulted in better visual outcomes at 1 and 2 years, particularly in those with worse visual acuity at baseline.

We haven't reported every pairwise comparison from all RCTs in this introductory paragraph. We aim to make the key point that all economic evaluations conducted alongside the RCTs have concluded that ranibizumab and aflibercept are not cost-effective when compared to bevacizumab.

3. Can you elaborate more on the methods for evaluating cost of ranibizumab and aflibercept? Does the NHS report directly state the cost, and how does one determine the cost of aflibercept or ranibizumab without the drug codes for each injection?

Response:

We have revised this section to clarify that the HPAI report does directly state the aggregate cost of ranibizumab and aflibercept. The hospital pharmacies who contribute data to the HPAI report have the drugs codes, hence the HPAI can estimate aggregate costs. However, drug codes are not recorded in hospital episode statistics (HES) meaning that analysis of anti-VEGF use by diagnosis or by region of England cannot be stratified by drug.

4. Page 9 lines 17-19 "Subsequent [...] bevacizumab," please cite and provide an example of the subsequent legal challenges and place this portion in the discussion.

Response:

We added a citation to a review article which provides details of one such legal challenge and have moved this portion to the discussion section.

5. Lines 9-11 on page 15 "The political decision [...] NHS patients" please cite some evidence of different nations support of bevacizumab use.

Response:

We added a citation that discusses the regulatory measures that other EU countries (Italy, France) and the WHO have taken to make bevacizumab more widely available for nAMD.

6. In the conclusion, this paper allows us to draw the conclusions that a. anti-VEGF drug use is increasing, b. costs of ranibizumab and aflibercept are increasing, c this is wide variation on use of these medications throughout the UK, and that substituting bevacizumab would likely lead to relative savings compared to the status quo. Relative savings can be achieved if NICE recommended use of bevacizumab, however as anti-VEGF use increases overall costs may still rise.

Response:

We agree with the reviewer and these are the messages we summarise in our statement of principal findings.

7. In the conclusion, lines 14-17 should be reworded. Although I agree, and readers should be able to "connect the dots," this study does not directly show how anti-VEGF costs limit care and how NICE is unable to provide equal access to cost effective care. I think it would be preferable that after point 6 is addressed to note that NICE's inability to recommend bevacizumab, a more cost effective medication with proven efficacy, may lead to the inability to achieve equal access. Further, I do think it is accurate that in this case they are failing to encourage cost effective care.

Response:

We have reworded this section to better 'link up' the implications of our findings.

8. The title should be changed to “the cost of turning a blind eye to unlicensed medicines: NICE recommendations on anti-vascular endothelial therapy.”

Response:

We have changed the title in response to this suggestion and the editorial policy.

Reviewer: 2

Reviewer Name: Andrew F. Smith

Institution and Country: Dept. of Ophthalmology, King's College London, London, UK Please state any competing interests: None

Overview:

This is an important piece of research in a complex area, namely the prevention of vision loss and blindness due to ARMD in the UK and the cost implications of using expensive licensed medicines versus cheaper off-label drugs with similar efficacy and ensuring access by patients to these medications.

The assumptions of the analysis, however, are somewhat limited and largely restricted to considerations of the impact on the NHS drug budget and inequalities in accessing anti-VEGF treatment across the UK.

In my view, both central issues have been handled rather simplistically and would benefit from further elucidation of the wider issues at play.

Proposed revisions:

In particular the authors may wish to consider the following revisions which should yield a more nuanced treatment of this complex topic.

1) The authors have relied solely on HES statistics to explore trends in the use of licensed and off label anti-VEGF agents across the UK. Given the difficulties in precise coding, it might be worth obtaining access to IMS pharmaceutical sales data for the UK to conduct a parallel analysis by drug names to see if the same quantities of both the licensed and off label anti-VEGF medicines were sold during the same time periods explored and roughly match the analysis of the HES data sets.

Response:

We do not rely solely on HES data. We use data from HES and from HPAI (which is based on data collected from hospital pharmacies by IMS Health). We observed similar temporal trends in these two independent data sources which provides reassurance about the validity of our analyses.

We do not think that additional sales data to conduct a parallel analysis would be a useful addition. Such data would need to record prescribing by indication (to exclude use of bevacizumab in oncology) and by area (MSOA or CCG). We are not aware of any pharmaceutical sales data that does this.

2) Moreover, an analysis of actual UK pharmaceutical sales data should show the total quantities of both the licensed and off label anti-VEGF drugs sold in different parts of the UK and CCGs regions.

Response:

For the reasons stated in our previous response, we do not think this is the case.

3) The authors may also conduct a short survey of CCGs about their purchasing of anti-VEGF agents to see if they received any discounts and whether those CCGs who extended discounts also accounted for higher numbers of anti-VEGF injections relative to CCGs paying full price.

Response:

CCGs are bound by “commercial in confidence” clauses in their contracts with the pharmaceutical companies. This makes getting useful data from surveys and even freedom of information requests impossible. It is possible that the CCGs who get the best discounts have the highest procedure rates. We have revised our manuscript (p13) to discuss this possibility.

4) No mention was made of perhaps the most important factor governing who has access to licensed or off-label anti-VEGF medicines, namely the supply and distribution of ophthalmologists as well as specially trained nurse practitioners across the UK. This could be achieved relatively easily by superimposing a map showing the distribution of ophthalmologists across the UK (particularly medical retinal specialists) and then correlating this distribution of eye care personnel to inequalities in access anti-VEGF agents. I suspect that accessing eye care for wet ARMD has much more to do with eye care personnel than the cost of the licensed anti-VEGF agents. The Royal College of Ophthalmologists would, I am sure, be able to assist with the provision of such workforce data. Alternatively, CCGs should have a list of the number of ophthalmologists by staff grade which could be used for such purposes.

Response:

We have expanded the section in our discussion that points out that observed “variations may be due to the availability of trained staff and facilities in local hospitals” to further emphasize this as a likely cause of some of the observed variation.

The reviewer is suggesting a substantial addition to our study which, we believe, would detract from the focus of our paper. We have demonstrated the size of the inequalities of access to therapy (which is the most relevant issue for patients). We have demonstrated that the inequalities occur within as well as between CCGs, suggesting that CCG policies and spending are not the only factors driving inequalities. The reviewer is suggesting an interesting analysis, but one that would be best done in a separate paper exploring the role of the many factors (e.g. travel distance, ophthalmologist capacity) that might cause inequalities.

5) The economic assumptions of the model make no allowance for the fact that the total societal costs of treating and managing patients with ARMD may well be on the decline, such that increased expenditures on licensed or off-label anti-VEGF agents owing to their efficacy, may actually be decreasing consumption on other healthcare or societal costs such as nursing home care, low vision rehabilitation services, and mobility aids, etc. for those who would have otherwise gone blind due to untreated wet ARMD. The authors might do well to calculate the cost of untreated vision loss and blindness due to wet ARMD versus the cost of treating wet ARMD patients either with the licensed or off-label anti-VEGF before concluding that the NHS is not getting value for money at least from a macro-economic or societal perspective.

Response:

We have expanded the section in our discussion to note that improved visual acuity will lead to longer term savings to the NHS and social care services.

However, calculating the cost of untreated vision loss, suggested by the reviewer, is not relevant to the comparison we are making between treating with ranibizumab/aflibercept or treating with bevacizumab (an unlicensed, cheaper drug of similar effectiveness). We are arguing that the Government refusal to support CCGs who want to use bevacizumab, which has been proven to be more cost-effective than ranibizumab and aflibercept, does mean that the NHS is not getting value for money from the NHS and societal perspective.

6) Another crucial point worth exploring is the degree to which those captured in the HES dataset represent “true” candidates and that ophthalmologists are not over treating ARMD patients and “cashing in” on the anti-VEGF ARMD bonanza, especially when the consequences of no treatment are further vision loss and blindness to patients. In essence, how does one know that all ARMD patients receiving either the licensed or off-label anti-VEGF treatment meet the criteria and are truly medically warranted? Or could ophthalmologists simply be over treating ARMD patients in the UK for financial gain? This could be answered by examining the number of persons in the UK who would have had the treatable wet form of ARMD from epidemiologic studies and comparing this with the numbers actually treated in the UK during the same time period to see if an appropriate number were being treated or not. This could be achieved by making assumptions about the number of injections per patient and examining the total number of injections annually during the period of interest and comparing this with the expected epidemiological burden of wet ARMD in the UK over the same time period. We would then know if the ophthalmologist were pulling the wool over the eyes of the NHS!

Response:

We are not aware of any UK evidence that ophthalmologists are over-treating patients for financial gain. The NICE criteria for treatment are objective, reducing the risk of overtreatment. We discuss the need for further evidence to make sure the balance between monitoring and treatment is right.

The substantial addition that the reviewer suggests is not practical (it would have to be done for nAMD, DMO, RVO) and would require too many assumptions to provide robust evidence on any over-use of anti-VEGF therapy.

7) In connection with the above point (6), the authors might wish to consider the provision of a better means of monitoring who would benefit from anti-VEGF agents (licensed or off-label) and thereby cut down on the potential for abuses in the system, including, expanding the indications beyond the realm of sufficient clinical data. This in turn, might well attenuate the rising costs.

Response:

We have revised a section of our discussion in order to recommend coding of high cost drugs in HES datasets in order to monitor use.

8) The authors further assume that the efficacy between Lucentis and Avastin are comparable. A review of reported adverse events may well reveal this not to be the case owing to basic pharmacological differences in the mechanism of action between the two anti-VEGF agents, with one having a larger molecular weight and longer half-life in the circulatory system (the off-label Avastin) versus that with the smaller molecular weight and shorter half-life (the licensed anti-VEGF agent (Lucentis)).

Response:

This is not an assumption, we have referenced the best evidence [references 6-13 in paper] that bevacizumab and the licensed drugs have similar efficacy in nAMD, DMO and RVO and a similar adverse event profile in nAMD.

9) Finally, the authors may wish to comment on the precedent that the Lucentis / Avastin saga has set for the role of Medicine Control Agencies, like the EMA and FDA whose primary concern remains the licensing of safe and effective medications for its populations, while concerns regarding costs and cost-effectiveness remain the preserve of other agencies, and seemingly secondary agencies, such as NICE in the UK or CMS in the USA for example.

Response:

As we state in our discussion, the main implication of our findings is the huge cost of the political decisions to narrow NICE's remit to exclude cheaper unlicensed drugs from their technology appraisals and guidelines and not to support commissioners who want to use cheaper unlicensed drugs of similar effectiveness.

VERSION 2 – REVIEW

REVIEWER	Ravi Parikh MD MPH USA
REVIEW RETURNED	02-Sep-2017

GENERAL COMMENTS	Despite limitations (such as specific drug codes) I think this provides important data for utilization rates in the UK
--

REVIEWER	Andrew F. Smith Dept of Ophthalmology Kings College London, London, UK
REVIEW RETURNED	28-Aug-2017

GENERAL COMMENTS	It would be interesting to overlay the distribution of ophthalmologists over the findings of this research to see how it correlates with access by patients to anti-VEGF across England.
--

VERSION 2 – AUTHOR RESPONSE

Reviewer 1

Comment: Despite limitations (such as specific drug codes) I think this provides important data for utilization rates in the UK

No response required

Reviewer 2

Comment: It would be interesting to overlay the distribution of ophthalmologists over the findings of this research to see how it correlates with access by patients to anti-VEGF across England.

Response: We considered this. However, as not all ophthalmologists provide anti-VEGF therapy, those that do may work across several hospitals, and those hospitals may provide outreach treatment clinics at several locations, we do not believe this would improve the paper.